# The Generation of Nitric Oxide from Aldehyde Dehydrogenase-2: The Role of Dietary Nitrates and Their Implication in Cardiovascular Disease Management

**DOI:** 10.3390/ijms232415454

**Published:** 2022-12-07

**Authors:** Jessica Maiuolo, Francesca Oppedisano, Cristina Carresi, Micaela Gliozzi, Vincenzo Musolino, Roberta Macrì, Federica Scarano, Annarita Coppoletta, Antonio Cardamone, Francesca Bosco, Rocco Mollace, Carolina Muscoli, Ernesto Palma, Vincenzo Mollace

**Affiliations:** 1Pharmaceutical Biology Laboratory, in Institute of Research for Food Safety & Health (IRC-FSH), Department of Health Sciences, University “Magna Graecia” of Catanzaro, 88100 Catanzaro, Italy; 2Institute of Research for Food Safety & Health (IRC-FSH), Department of Health Sciences, University “Magna Graecia” of Catanzaro, 88100 Catanzaro, Italy; 3Renato Dulbecco Institute, Lamezia Terme, 88046 Catanzaro, Italy

**Keywords:** aldehyde dehydrogenase 2 (ALDH2), nitrate tolerance, nitric oxide (NO), endothelium, oxidative stress, nitrate–nitrite intake

## Abstract

Reduced bioavailability of the nitric oxide (NO) signaling molecule has been associated with the onset of cardiovascular disease. One of the better-known and effective therapies for cardiovascular disorders is the use of organic nitrates, such as glyceryl trinitrate (GTN), which increases the concentration of NO. Unfortunately, chronic use of this therapy can induce a phenomenon known as “nitrate tolerance”, which is defined as the loss of hemodynamic effects and a reduction in therapeutic effects. As such, a higher dosage of GTN is required in order to achieve the same vasodilatory and antiplatelet effects. Mitochondrial aldehyde dehydrogenase 2 (ALDH2) is a cardioprotective enzyme that catalyzes the bio-activation of GTN to NO. Nitrate tolerance is accompanied by an increase in oxidative stress, endothelial dysfunction, and sympathetic activation, as well as a loss of the catalytic activity of ALDH2 itself. On the basis of current knowledge, nitrate intake in the diet would guarantee a concentration of NO such as to avoid (or at least reduce) treatment with GTN and the consequent onset of nitrate tolerance in the course of cardiovascular diseases, so as not to make necessary the increase in GTN concentrations and the possible inhibition/alteration of ALDH2, which aggravates the problem of a positive feedback mechanism. Therefore, the purpose of this review is to summarize data relating to the introduction into the diet of some natural products that could assist pharmacological therapy in order to provide the NO necessary to reduce the intake of GTN and the phenomenon of nitrate tolerance and to ensure the correct catalytic activity of ALDH2.

## 1. Introduction

The treatment of cardiovascular diseases involves the use of organic nitrates such as glyceryl trinitrate (GTN). Its clinical effect is delivered via the dilation of large conductance veins and large arteries, as well as by the inhibition of platelet aggregation [1]. At the level of vascular smooth muscle cells (SMCs) and platelets, this action involves—first of all—the bio-activation of GTN and other nitrates in order to release nitric oxide (NO) or S-nitrosothiol, together with high levels of cGMP (cyclic guanosine monophosphate) [1,2]. However, the chronic use of organic nitrates generates a phenomenon known as “nitrate tolerance” [1,3], which could be reduced by intermittent administration—but such an approach would reduce the therapeutic effect [1]. Furthermore, it has been shown that such tolerance is accompanied by an increase in oxidative stress, endothelial dysfunction, and sympathetic activation [1]. Which pathophysiological events are responsible for these conditions is still not very clear; however, what can be determined with more certainty is the involvement of the mitochondrial isoform of aldehyde dehydrogenase 2 (ALDH2), which is an enzyme that catalyzes the bio-activation of GTN to NO [1,4,5]. The latter activates guanylate cyclase (GC)-cGMP-protein kinase G (PKG) signal transduction, which leads to the vasodilation and inhibition of platelet aggregation [1,6]. Vasodilation involves both the arterial and venous systems. In particular, at the level of the arterial system, the role of GTN is to reduce the afterload, dilate the coronary arteries, and prevent vasospasms. At the level of the venous system, GTN vasodilates the veins, reducing the preload. In doing so, GTN has a dual hemodynamic action, thereby allowing the heart to have the right balance between oxygen consumption and supply [6]. In regard to the ischemic heart, GTN is able to restore the balance between the supply and demand of oxygen and nutrients [7]. Moreover, it has also been shown that the catalytic activity of ALDH2 is reduced in the presence of a free radical species [1,8]. In particular, the NO released by the reaction catalyzed by ALDH2 reacts with the superoxide and causes the formation of peroxynitrite, which could lead to nitration of the ALDH2, as well as causing a loss of its catalytic activity and the subsequent generation of nitrate tolerance [1,9]. As a consequence of this tolerance, a higher dosage of GTN is necessary in order to obtain the same vasodilatory and antiplatelet effects [1]. Due to the fact that continuous treatment with GTN can lead to nitrate tolerance, it would be preferable to administer natural products that can modulate the concentration of NO without developing side effects. It has been amply demonstrated that proper nutrition protects human health from various pathologies. In addition, the consistent intake of fruits and vegetables can increase NO levels. Therefore, the purpose of this review is to describe the main products of plant origin that can be used to increase NO in assistance of drug therapy, in order to reduce GTN administration and the consequent onset of nitrate tolerance and maintain the unaltered catalytic activity of ALDH2. This need derives from a study of the literature present in PubMed, which has shown that in the last 10 years there has not been a review that addresses this fundamental topic for cardiovascular disease therapy, considering both the mechanisms involved and the possible natural products present in the diet.

## 2. Vascular Endothelium and Endothelial Dysfunction 

The endothelium is a cell monolayer composed of approximately 10^13^ endothelial cells (ECs). This layer covers the inner surface of the blood vessels, lymphatic vessels, and heart. In relation to the mesenchymal derivation, the endothelium is the largest tissue in the body, with a total weight of around 1.0–1.8 kg. Furthermore, its cells represent about 1.5% of total body mass [10]. Until the early 1980s, the endothelium of the vascular tree was believed to play a passive role in forming the shell of the vascular shaft, dealing only with selective permeability to water and electrolytes. Today, it is known that the endothelium carries out numerous functions, including regulation of the tone and the structure of the vessels [11]; regulation of vasal permeability [12]; angiogenesis [13]; hemostasis checks [14]; control of inflammation and recruitment of neutrophils [15]; and endocrine–metabolic functions [16]. It is possible to define the endothelium as an endocrine, paracrine, and autocrine organ, which is capable of releasing a wide variety of substances in the blood and interstitial space (including vasoactive compounds, growth factors, inflammation mediators, adhesion molecules, hemostatic system proteins, and extracellular molecules). These substances can act remotely (via endocrine activity), on nearby cells (via paracrine activity), or on the cell that produced them (thereby demonstrating an autocrine activity); it is these activities that are responsible for the balance maintained by the endothelium [17]. In particular, blood flow is regulated through the secretion and absorption of vasoactive substances by the endothelium, which act in a paracrine fashion to shrink and dilate specific vascular beds [18,19]. When the endothelium is working normally, all the appointed functions are carried out adequately, which consequently also involves a proper immune response. On the other hand, endothelial dysfunction characterized by reduced vasodilation—which is a pro-inflammatory state and when active pro-thrombic properties are present—is associated with most forms of cardiovascular diseases, such as coronary heart disease, hypertension, diabetes, chronic kidney failure, peripheral vascular disease, and severe viral infections [20,21]. Endothelial dysfunction is involved in other regions [22,23] and also causes considerable damage to the nervous system [24,25,26,27]. Vascular tone is defined by the balance between the degree of constriction of the blood vessel and its maximum dilation; in addition, it is modulated by the release of relaxing as well as constrictive factors derived from the endothelium. In fact, ECs physiologically synthesize and release several relaxing factors derived from the endothelium—including vasodilator prostaglandins, endothelium-dependent hyperpolarization factors, and NO, but also contraction factors such as thromboxane A2, endothelin, angiotensin II, superoxide, and the correct balance between the production of vasodilators and vasoconstrictors (which ensures proper maintenance of vascular tone) [10]. NO is a soluble gas that demonstrates important vaso-relaxant protective functions and is regulated by nitric oxide endothelial synthase (e-NOS), which is an enzymatic isoform constitutively expressed in ECs. In particular, this enzyme catalyzes the conversion of L-arginine to L-citrulline and NO. When NO is synthesized, it spreads into smooth vascular muscle cells, stimulating soluble guanylate cyclase and the increasing of cyclic guanosine monophosphate (which is an NO effector that promotes vasodilation) [28]. Reactive oxygen species (ROS) are reactive oxygen intermediates that form physiologically as byproducts of cell metabolism. When present at physiological concentrations, ROS are very useful for cellular homeostasis, acting as second messengers in the transduction of cellular signals and predisposing toxicity reactions against bacterial infections [29]. On the other hand, when ROS levels exceed the antioxidant capacities of the cell or when the antioxidant enzymes have reduced activity, the onset of oxidative stress occurs. This condition is extremely dangerous, as ROS can react with major biological macromolecules and alter them accordingly [30,31]. Cell membranes are particularly susceptible to oxidative damage caused by ROS and can encounter “lipid peroxidation”, a process in which ROS remove electrons from lipids and damage phospholipids. This alteration can also lead the cell to apoptotic death [32,33]. ROS accumulation is involved in the onset of several diseases, including cancer, as well as many metabolic diseases such as diabetes and obesity, neurodegenerative disorders, lung diseases, and kidney diseases [34,35,36,37], among others. It has recently been shown that there is a close correlation between accumulation of ROS and increased inflammation and endothelial dysfunction [38,39,40]. For example, the reduction in bioavailability of NO can occur not only for decreased e-NOS protein expression, but also as a result of an increased level of ROS and, especially, of superoxide anions (O^•−^), which are responsible for the formation of peroxynitrite (ONOO^−^). The latter promotes protein nutrition by contributing to endothelial dysfunction and cellular death [41,42]. In most instances, the physiological antioxidant mechanisms of the human body are able to neutralize ROS [43]. However, in their absence endothelium lesions, relative dysfunctions, and alterations in the content of NO can occur [44,45]. Endothelial dysfunction—which is related to the production and accumulation of ROS, as well as reactive nitrogen species (RNS)—is particularly involved in some pathologies, such as hypertension, hyperlipidemia, atherosclerosis, and diabetes mellitus, which all have connections with vascular damage as a common denominator [46]; in fact, endothelial dysfunction can be considered an early marker of cardiovascular events [47,48]. In smooth muscles, the listed pathologies or bad habits—including smoking and alcohol intake—can lead to activation of the enzyme NADPH oxidase, which is responsible for the desensitization of soluble guanyl cyclase and the breakdown of cyclic guanosine monophosphate [49]. The close correlation between endothelial damage and cardiovascular risk is also associated with decreased NO bioavailability, as well as impaired activity of endothelial NO synthase [50,51]. Today, it is known that endothelial dysfunction also involves the alteration of ALDH2 expression, thereby resulting in the alteration of the oxidative state and the onset of inflammation [52]. Due to the fact that, as already described, ALDH2 protects endothelial cells from oxidative stress, these cells are thus able to turn their physiological function on or off [53,54]. A representation of this double control of the endothelium regarding ALDH2 is shown in Figure 1.

### Nitrates, Cardiovascular Diseases, and Tolerance

Reduced bioavailability of the NO signaling molecule due to its reduced synthesis or excessive consumption has been associated with the occurrence of cardiovascular diseases. For this reason, its restoration guarantees a mechanism that has a positive effect on these pathologies [55]. In addition, it is important to act on the reduced availability of substrates and cofactors, on the generation of ROS, and on the oxidation of NOS, which are factors that disturb the signaling NO and accentuate the development and pathogenesis of cardiovascular diseases [56]. A highly interesting source of indirect syntheses of NOs is provided by the inorganic anions nitrate (NO_3_^−^) and nitrite (NO_2_^−^). In fact, nitrates and nitrites are physiologically transformed in the blood and tissues to form NO and other oxides of bioactive nitrogen; moreover, they should be considered a reserve of NO. Nitrates are transformed into nitrites in the gastrointestinal tract. This is achieved through the reducing power of some intestinal bacteria. On the other hand, nitrites are reduced to NO in the oral cavity and stomach, where the pH is particularly acidic. In this way, the nitrate–nitrite–NO path becomes complete [57]. The trinomial nitrate–nitrite–NO is fundamental in restoring the health of the cardiovascular system and generating NO-similar protective effects. This is in addition to the fact that it is widely used as a treatment in myocardial infarction, hypertension, and peripheral artery diseases [58]. The hypothesis that nitrates could produce an effect similar to that of NO was confirmed in 2006 by a research group who demonstrated that the administration of sodium nitrate for 3 days was able to reduce blood pressure in young, healthy individuals [59]. Subsequently, these protective effects of nitrates were confirmed in animal and human models of hypertension, oxidative stress, ischemia-reperfusion, and tolerance to hypoxia [60]. At the same time, the effects of nitrites on cardiovascular function were also examined [61], and with the same protective effect as NO, the nitrate–nitrite–NO pathway has become a therapeutic opportunity to be exploited in the event of cardiovascular disorders [62]. For example, Stokes et al. showed that dietary nitrite intake in mice with endothelial dysfunction (which was induced by dietary hypercholesterolemia) was able to preserve endothelial function, inhibit microvascular inflammation, and reduce increased expression of the C-reactive protein [63]. Another group found that nitrite intake reduced arterial stiffening and oxidative stress in a mouse model of endothelial dysfunction related to aging [64]. Moreover, sodium nitrate ingestion was able to mitigate endothelial function on ischemia-induced effects [65], reduce blood pressure [66], and attenuate cardiac hypertrophy and fibrosis in chronic models of hypertension [67,68]. Webb et al. showed, in an ex vivo mouse model of myocardial infarction, that nitrite could reduce the damaged area via a mechanism that involved the formation of NO [69]. Having said this, not only the heart was affected, but also other organs including the brain [70], kidneys [71], liver [72], and the hind limbs [73]. The mechanism of cytoprotective action carried out by nitrite appears to be the inhibition of mitochondrial respiration, with consequent reduction in the formation of ROS [74,75]. The encouraging effects of nitrates and nitrites as substrates for an in vivo formation of NO and its related oxides of nitrogen bioactives have stimulated the study of these clinical compounds in animal models; in particular, the beneficial effects justified considering these molecules as an opportunity for the development of drugs against cardiovascular and metabolic diseases. For this reason, phase II studies have been initiated on a model of acute myocardial infarction [76]. The use of organic nitrite was replaced by an organic nitrate, GTN, which was easier to administer and had a longer duration of action. GTN has been used in the treatment of angina pectoris and heart failure for over 150 years. The use of GTN results in the release of NO, the activation of guanyl cyclase, and the relaxation of blood vessels [77]. Although the protective effects of organic nitrates on cardiovascular dysfunction are undisputed, it is also known that chronic therapy with these compounds leads to long-term multifactorial effects that can be summarized as the development of a tolerance to nitrates; the onset of profound changes in vascular homeostasis with oxidative stress; and a suspicion that organic nitrate therapy may be associated with an increase in coronary events and neurohumoral adaptations [78]. Nitrate tolerance is defined as the loss of the hemodynamic effects of organic nitrates and the need for higher dosages in order to maintain the same effects. It is the consequence of a series of phenomena, both vascular and extravascular, which can generate important clinical implications [79]. In the development of nitrate tolerance, the role of mitochondrial oxidative stress was described in a mouse model that had a heterozygous deletion of the enzyme manganese superoxide dismutase (MnSOD), which is the mitochondrial isoform of this enzyme. In addition, ROS has been shown to be responsible for disrupting cross-talk to the cytoplasm and mitochondria [80]. Interestingly, nitrates are responsible for inhibiting ALDH2, which is responsible for decreasing or canceling the protective mechanism against oxidative stress [81]. GTN can potently and rapidly inactivate ALDH2, which is an even earlier effect of nitrate tolerance. It was shown, for example, that in knockout mice for ALDH2, nitrate tolerance occurs more easily [82]. The severity of the tolerance is responsible for several effects, as is represented in Figure 2. Due to the discomfort of long-term administration of nitrates, it is essential to select a storage form of NO that may have an important therapeutic potential and advantages over organic nitrates.

## 3. The Role of ALDH2 

ALDH2 is considered a cardioprotective enzyme [83,84,85] that is capable of both preventing the onset of ischemic damage (during myocardial infarction) and also in reducing the infarct area [83]. In humans, the aldehyde dehydrogenase (ALDH) superfamily consists of nineteen NAD(P)^+^-dependent isozymes. These catalyze the oxidation of both exogenous (such as alcohol) and endogenous (such as lipids and amino acids) aldehydes into carboxylic acids [86,87,88,89,90,91]. This protects the cells from damage caused by active aldehydes and plays an important role in ROS elimination. In fact, high concentrations of non-metabolized aldehydes cause enzyme inactivation, DNA damage, and cell death [88]. Among these, ALDH2 is present in various organs, including the heart; in particular, it is localized at the level of mitochondria, organelles that are important for ROS and reactive aldehyde generation [92,93]. It is a tetrameric allosteric enzyme involved in the metabolism of ethanol. The latter is first converted to acetaldehyde in a reaction catalyzed by alcohol dehydrogenase (ADH) and then forms acetic acid in a reaction catalyzed by ALDH2 [92]. In addition to its dehydrogenase activity, ALDH2 also possesses esterase and reductase activity. Furthermore, its role in GTN-induced vasodilation is linked to its reductase activity [7]. The mechanism by which ALDH2 catalyzes the bioactivation of GTN to NO has been studied in recent years by various research groups. In particular, Lang et al. (2012), by means of crystallography and mass spectrometry studies conducted on wild-type ALDH2 and on the triple mutant of the protein with reduced denitration activity (E268Q/C301S/C303S), tried to elucidate this mechanism. Indeed, it has been established that the denitration of GTN begins with the nucleophilic attack of Cys-302, present in the catalytic site of the enzyme, on a terminal nitrogen of GTN. This results in the formation of a thionitrate adduct as the first reaction intermediate and the release of 1,2-glyceryl dinitrate (1,2-GDN). At this point, the intermediate can undergo nucleophilic attack by the flanking cysteines, Cys-301 or Cys-303, with the formation of nitrites and a disulfide bond in the active site of the enzyme, which would lead to the reversible inhibition of ALDH2. The released nitrite would subsequently be reduced to NO, or ALDH2 could be irreversibly inhibited with the formation of sulfinic acid in the active site, mediated by the presence of Glu-268. This pathway could play a fundamental role in the development of nitrate tolerance. A third pathway leads to the direct production of NO and the reversible inhibition of ALDH2 [94].

### 3.1. Oxidative Stress, Toxic Aldehydes, and ALDH2 

Certain studies conducted in vivo by performing pretreatments with nitroglycerin for 8 days (as well as in vitro studies on isolated rat thoracic aortas and on human umbilical vein endothelial cells (HUVECs) treated with nitroglycerin) have shown that, during the nitrate tolerance process, an increased ROS formation reduces ALDH2 activity. This phenomenon results in a reduction in the release of the endogenous calcitonin gene-related peptide (CGRP) [95]. Hence, chronic nitroglycerin treatment results in an increase in mitochondrial and vascular oxidative stress, as well as a reduction in ALDH2 activity in the aorta and cardiac mitochondria at the same time [96]. In fact, during heart failure (HF), ROS and oxidative stress play a fundamental role both in the phase of myocardial remodeling and in the condition of overt HF. It is known that the intracellular ROS sources are due to the activities of NADPH oxidase (NOX), an enzyme that with its two isoforms NOX1 and NOX4 generates superoxide, clinically associated with atherosclerosis and therefore cardiovascular disorders [97], xanthine oxidase, and nitric oxide synthase [98]. This is in addition to the activity of an enzyme localized in the external mitochondrial membrane, monoamine oxidase (MAO)—of which there are two isoforms, MAO-A and -B. MAO-catalyzed oxidative deamination reactions produce hydrogen peroxide and aldehydes, which can be eliminated by ALDH2 [98]. A reduction (or an inhibition) in ALDH2 activity leads to the accumulation of toxic aldehydes generated by MAO, thereby causing mitochondrial dysfunction and consequent myocardial failure. This supports the fundamental role of ALDH2 in eliminating toxic aldehydes in the myocardium in order to protect the heart from oxidative stress [98]. It is reported that in the vascular endothelium, the loss of ALDH2 activity leads to endothelial dysfunction, as there is an increase in ROS levels, an accumulation of 4-hydroxy-2-nonenal (4-HNE) protein adducts, and a loss of mitochondrial bioenergetic functions. Therefore, senescence of the endothelium with loss of regenerative capacity can be a defensive response to the damage caused by the accumulation of toxic aldehydes, thus leading to a loss of function in the vascular system [53]. Furthermore, in a study conducted on cell model RAW264.7 that was treated with oxidized low-density lipoprotein (ox-LDL), it was determined that ALDH2 activation reduces ox-LDL-induced 4-HNE production. Therefore, oxidative stress in atherosclerosis could be reduced through the inhibition of activation of the NLRP3 inflammasome by ALDH2, making it a potential target for anti-inflammatory therapies [99,100].

### 3.2. ALDH2 and Ischemia-Reperfusion Injury (IRI) 

It is known that ALDH2 has a cardioprotective function, which also limits ischemia-reperfusion injury (IRI) [92,93,101,102]. This condition arises as a consequence of reperfusion, which is a necessary treatment to reduce the magnitude of myocardial infarction, as well as the direct manifestation of coronary artery disease (CAD). IRI is also generated following cardiac arrest due to open-heart surgery with cardiopulmonary bypass (CPB), in which there is an increase in the synthesis of superoxide, ROS, and aldehydes. ALDH2 activation has been reported to reduce ischemic cardiac damage. Furthermore, there are various mechanisms underlying the cardioprotective effects of ALDH2. Among these, ALDH2 acts downstream of protein kinase C type ε (εPKC). In particular, it reduces the production of reactive aldehydes, such as 4-HNE, that are produced during the peroxidation lipid process. The cardioprotective effect of ALDH2, through its action on reactive aldehydes, has positive consequences in regard to ROS production, mitochondrial kATP channel regulation, and mitochondrial permeability transition pore (MPTP) openings [92,93,103,104,105]. In addition, the apoptotic process in cardiomyocytes also increases during IRI following an increase in ROS production. ALDH2, by reducing ROS production, reduces the activation of the JNK signaling pathway and the expression of c-Jun [92,93]. The autophagic process also increases during IR. Furthermore, ALDH2 protects cardiomyocytes by activating the LKB1/AMPK/mTOR pathway during the ischemic phase, as well as by acting on the PTEN/Akt/mTOR pathway during the reperfusion process, which in turn reduces excessive autophagy. Another cardioprotective mechanism of ALDH2 during IR is carried out through the reduction in the formation of reactive carbonyl species (RCS) that causes the carbonylation of proteins, with a consequent loss of activity [92]. In addition to this, in patients with CAD, ALDH2 plays a fundamental role in nitroglycerin metabolism, specifically for its bioconversion into NO in order to obtain a vasodilating effect [92]. It is known that after myocardial infarction, the co-administration of nitroglycerin and Alda-1—an activator of ALDH2—has positive effects on the metabolism of reactive aldehyde adducts by reducing the cardiac dysfunction generated by the use of nitroglycerin [7]. Indeed, in an animal model of IR, it has been shown that Alda-1, when administered before prolonged treatment with GTN, protects against heart damage from GTN-induced ALDH2 inactivation [106].

### 3.3. ALDH2 Polymorphism 

In the liver, alcohol that is consumed is metabolized into acetaldehyde in a reaction catalyzed by ADH. In the subsequent reaction catalyzed by ALDH, acetaldehyde is converted to acetate. The latter leaves the liver and is metabolized in the heart and muscles. As an aside, acetaldehyde is a very toxic intermediate. Moreover, there are two isoforms of ALDH; one is cytosolic and encoded by the *ALDH1* gene and the other is mitochondrial and encoded by the *ALDH2* gene. The mitochondrial isoform is very important to this process due to its great affinity for acetaldehyde (Km = 0.20 μM) [107,108]. The polymorphism of the *ALDH2* gene is linked to the role that the enzyme plays in the oxidation of alcohol. In fact, in exon 12, there is a single nucleotide polymorphism (SNP) that determines the substitution, at residue 504, of glutamic acid with lysine. The first condition corresponds to the *ALDH2*1* allele, while the 504 Lys allele is referred to as *ALDH2*2* and gives rise to a less active isozyme with a reduced ability to eliminate acetaldehyde. In regard to the people carrying this allele, the accumulation of acetaldehyde after alcohol consumption generates flushing, nausea, or vomiting, thereby increasing the risk of developing alcohol-related diseases [107]. The two isozymes are present in the Caucasian population, while about 30–50% of the East Asian population inherited the mutant *ALDH2*2* allele [109]. ALDH2 is considered the most important enzyme for GTN bioactivation, as it activates GTN at clinically relevant plasma concentrations, i.e., below 1 μM. Despite this, forearm blood flow (FBF) responses to a brachial artery infusion of nitroglycerin in subjects with and without ALDH2 Glu504Lys polymorphism showed that ALDH2 is not the only enzyme responsible for the bioactivation of nitroglycerin at therapeutically relevant or higher concentrations. Studies of ALDH2 Glu504Lys polymorphism are important, as Glu504Lys is a common genetic variant that greatly reduces ALDH2 activity [110]. Taking into account that the enzymatic activities of the wild type (Glu504, encoded by *ALDH2*1*) and mutant (Lys504, encoded by *ALDH2*2*) proteins vary as a function of the functional genetic polymorphism of the protein [6,111], Miura et al. conducted a study on the vasodilator effect of GTN sublingual tablets administered to three different Japanese genotypic groups (*ALDH2*1/*1, ALDH2*1/*2,* and *ALDH2*2/*2*). The results of in vivo vasodilation determined by sublingual GTN did not show differences between the genotypes, even with respect to the degree of vasodilation. Having said this, the enzymatic activity of ALDH2 was different between the various groups studied, thus indicating the presence of other pathways aside from that of ALDH2 for the purposes of nitroglycerin bioactivation [6]. In addition, a study was conducted on patients with coronary spastic angina (CSA) in order to determine any differences in the response of nitroglycerin-mediated dilation (NMD) and flow-mediated dilation (FMD) between the wild type *ALDH2*1/*1* and mutant *ALDH2*2* (E487K point mutation). The results obtained by Mizuno et al. demonstrated that all patients reported comparable endothelial dysfunction as well as nitrate tolerance following continued GTN administration for 48 h. In patients with the *ALDH2*2* mutation, however, tolerance was more severe, with a lower response to GTN at baseline [112]. Thus, treatment of coronary heart disease with GTN in *ALDH2*2* subjects is clinically ineffective. Therefore, a reduced ALDH2 activity determines an inefficient elimination of reactive aldehydes with a consequent increase in cytotoxicity and oxidative stress [113]. In *ALDH2*2* mutant mice, the administration of empagliflozin (EMP), a sodium–glucose cotransporter (SGLT) 2 inhibitor, has been shown to reduce the onset of diabetic cardiomyopathy via limiting the formation of 4HNE protein adducts, due to the condition of hyperglycemia. This improvement in cardiomyopathy occurs despite the mutant mice possessing low ALDH2 activity. The action of EMP is also reported in diabetic patients with the same mutation [114]. A study conducted in *ALDH2*2* mutant mice demonstrated that coenzyme Q10—in addition to improving mitochondrial oxidative stress and preserving bioenergetics—is effective in protecting against attacks of atrial fibrillation (AF). The multi-omics studies conducted have made it possible to establish that coenzyme Q10 could be administered in humans who are characterized by the *ALDH2*2* genotype in order to ensure protection from AF attacks [115].

## 4. Treatment of Nitrate Tolerance

Experiments conducted on rat aortas have shown that treatment with serelaxin and low-dose GTN reduces GTN-induced tolerance. Serelaxin represents a new vasoprotective peptide capable of reducing oxidative stress and improving endothelial function. In particular, within the study conducted by Leo et al., it is reported that serelaxin co-treatment with low-dose GTN reduces superoxide production and increases ALDH2 expression. Most likely, serelaxin acts on the signaling mechanism of the eNOS pathway by regulating the expression of *Dhfr* [116]. For some years it has seemed that the onset of nitrate tolerance could be overcome by the use of co-therapies with antioxidant compounds rather than being overcome with therapeutic plans that foresee a “nitrate-free interval” [117]. In this context, the use of antioxidants—such as Mn (III) tetrakis (4-Benzoic acid) porphyrin (MnTBAP)—antagonize the development of nitrate tolerance as they reduce ALDH2 nitration and restore NO production as well as the consequent therapeutic effects on cardiovascular diseases [1]. Furthermore, it has been shown that nitrate tolerance, due to oxidation of the sulfhydryl groups present in the active site of ALDH2 (specifically the oxidation of cysteine 302, which is the critical catalytic amino acid of ALDH2 [106]), may not develop due to the pre-administration of lipoic acid, which can prevent or modify the oxidative inhibition of the enzyme. In fact, within the mitochondria, lipoic acid is enzymatically reduced to dihydrolipoic acid, which is capable of reducing the disulfides at the ALDH2 active site [118]. Furthermore, in vitro studies conducted on rat aorta rings have shown that freeze-dried (dealcoholized) red wine (FDRW), due to its polyphenol content, may be partially effective against GTN tolerance. The FDRW was used at a concentration equal to that reached by the total polyphenols in the plasma after drinking 100 mL of red wine. Moreover, its effect on nitrate tolerance is probably linked to the antioxidant properties of and the action on the endothelial function of the polyphenols present in it. In particular, the mechanism does appear to be mediated by superoxide dismutase (SOD) [119,120]. As such, in order to clarify the link between oxidative stress and the development of tolerance to organic nitrates, which is as a consequence of the administration of GTN, ex vivo experiments were conducted on rat thoracic aortas. This was conducted in order to test new synthesized organic nitrates with and without antioxidant properties. The results obtained showed that the vasodilation and tolerance profiles of the tested molecules varied according to their ability to interact with ALDH2 (which is responsible for their bioactivation), thereby confirming the fundamental role of ALDH2 in the development of nitrate tolerance. Another important result was that of relating the nitrooxy derivatives to possessing antioxidant properties. In fact, despite the fact that they were bioactivated by ALDH2, they did not inactivate the enzyme, thus demonstrating the involvement of ROS in deactivating ALDH2 [81]. Another way to prevent and modify GTN tolerance, as well as its subsequent endothelial dysfunction, could be a change in ALDH2 expression. In this sense, a study was conducted in HUVECs that were transfected with the ALDH2 gene, thereby inducing GTN tolerance. The results obtained demonstrated that the overexpression of ALDH2 protects cells from death, due to the fact that it reduces GTN-induced cytotoxicity and the oxidative damage that results from nitrate tolerance. In addition, lower ROS production and lower heme oxygenase 1 expression have also been reported [121].

## 5. Nitrates–Nitrites–Nitric Oxide in Fruits and Vegetables

Nitrogen is a fundamental chemical element necessary to ensure the survival of living organisms. In fact, it not only constitutes 78% of the atmosphere, but is contained in the main macromolecules necessary to ensure life on earth: DNA, RNA, and proteins. Nucleic acids (DNA and RNA) are macromolecules formed by long nucleotide chains, in which nitrogen and phosphorus are particularly relevant; in particular, DNA depends on the order in which the four nitrogenous bases, adenine (A), thymine (T), cytosine (C), and guanine (G) are arranged. Proteins are also polymers: individual monomers are made of nitrogen, carbon, hydrogen, and oxygen [122]. However, nitrogen would not be available without some important chemical processes that transform atmospheric nitrogen into organic compounds. These chemical transformations are guaranteed by the nitrogen cycle, a “gaseous” biogeochemical cycle whose main reservoir is the atmosphere. Atmospheric nitrogen, whose molecular formula is N_2_, is formed by two nitrogen atoms strongly bound together to generate an inert gas, that is, a poorly reactive compound that hardly reacts with the surrounding environment. Because of this inertia, nitrogen is difficult to assimilate in the world of the living, making its organication necessary. In this process, known as “nitrogen fixation”, molecular nitrogen (N_2_) is split and made available for living beings who convert it into oxides of nitrogen (NO_x_) and ammonia (NH_3_). A subsequent process known as “ammonification”, carried out by fungi and various soil bacteria, is responsible for the acquisition of a proton by the ammonia, which generates the ammonium ion (NH_4_^+^). The latter can be nitrified by some bacteria free in the soil; initially, nitrosanct bacteria will transform ammonium into nitrite; later, nitrifying bacteria will transform nitrites into nitrates. Nitrates represent the most bioavailable form of nitrogen for plants: in fact, they are absorbed at a radical level, transformed into vegetable proteins and used as a source of nitrogen for the whole trophic network. The nitrogen cycle ends with the “denitrification” phase where some bacterial species perform an anaerobic respiration, in which nitrate represents the electron acceptor instead of oxygen, leading again to the formation of bimolecular nitrogen [123]. In Figure 3, a representation of the biogeochemical nitrogen cycle is represented.

The reduction of nitrites and nitrates in NO has been discovered in the last decade and is becoming increasingly important, as it is involved in the prevention or reduction of cardiovascular, metabolic, and muscle disorders associated with decreasing NO levels. For this reason, it is important to estimate the amount of NO and its metabolites in different bodily regions, such as blood, urine, fluids, and various tissues [124]. Among them, blood, thanks to its easy accessibility, is the component chosen for the estimation of NO metabolites. Concentrations of nitrites in the blood (and in most organs and tissues) are represented in a low nanomolar or micromolar range. Nitrate is usually present in much higher amounts—in the micromolar range [125,126]. To date, the methodologies used to quantify NO and its metabolites, nitrates and nitrites, in various biological samples are multiple, although methods based on the Griess reaction, originally described in 1879, remain the most reliable. However, despite modern modifications, the limit is always in the sensitivity of the test [127,128]. Cardiovascular disorders are also related to numerous bad habits, such as smoking; excessive alcohol consumption; excessive body weight; a sedentary lifestyle; a low intake of fruits and vegetables; excessive sodium intake; lack of certain vitamins (folic acid, riboflavin, and vitamins C and D); and concomitant pathological forms (diabetes, hypertension, hypercholesterolemia, hypertriglyceridemia, and cardiac dysfunction) [129,130,131]. In addition, cardiovascular disorders are associated with endothelial dysfunction and a vascular endothelium alteration that precedes the development of cardiovascular events and promotes pathological amplification [132]. No effect produced by endothelium and nitrates–nitrites is capable of generating NO, as has already been widely described. Moreover, a substantial effect on the maintenance of vascular homeostasis, thanks to its powerful dilator effect, is found in the substantial delay of atherogenesis, blood pressure control, etc. [133]. One limitation, however, is provided by the knowledge that the chronic exogenous intake of nitrate–nitrites–NO inhibits the onset of tolerance. For this reason, it has been hypothesized in recent years that vegetable intake—which can release NO—could reduce the incidence of cardiovascular disease [134]. A diet rich in fruits and vegetables has increased interest in so-called “functional foods” and their application in health and disease [135]. The current acceptable daily intake for nitrites is 0.07 mg per kilogram of body weight per day (mg/kg body weight/day), while for nitrates it is 3.7 mg/kg body weight/day [136]. If these doses are exceeded, toxic nitrosamines can be generated; furthermore, these compounds are inhibited by antioxidant substances when the molar ratio of antioxidants/nitrites is greater than 2:1 [137]. The most nitrate-rich foods are vegetables and fruits (81–83%), due to the fact that plants accumulate these compounds easily. The factors contributing to a high nitrate content in vegetables are plant treatment with fertilizers, the growing conditions of the plant, the humidity level of the soil, the amount of rain, the intensity of light, nitrate reductase activity, seasonality, and cultivation systems, among others [138]. It has been shown that heat treatments, storage conditions, and some production processes (acidification, pasteurization, brining, and the shelf-stable process) are responsible for reducing nitrate content. For this reason, it is preferable and recommended that there is a greater consumption of fresh products [139]. The transformation of nitrates, taken with food, is represented in Figure 4.

It is worth remembering, however, that an indiscriminate increase in nitrates/nitrites through nutrition leads to systemic damage. The toxic action of nitrates and nitrites results mainly in metemoglobinemia (more frequently in children) and nitrosation of amines responsible for the onset of carcinogenic effects. When these compounds bind with hemoglobin, they form methemoglobin: in this case, the hemoglobin is unable to release oxygen effectively to body tissues, thereby inducing serious damage to the body up to death. Children are particularly affected by methemoglobinemia when ingesting water contaminated with large levels of nitrates [140]. A preventive strategy could be to check the concentration of nitrates in the water used in advance and thus avoid the consumption of vegetables, which are rich in nitrates, up to 4–6 months of age. Subsequently, in adult nutrition, the concentration of nitrate–nitrites is particularly high in meat and processed meat, in which these compounds are used as substances capable of protecting foods against the deadly bacteria *Clostridium botulinum* [141]. Often, the nitrites that are ingested, or formed as a result of nitrate reduction, can react with amines and amides. Furthermore, they can generate highly carcinogenic N-nitroso compounds [142]. As the transformation of dietary nitrates and nitrites to nitric oxide has beneficial effects in cardiovascular diseases, and, at the same time, due to the fact that high concentrations are toxic, it is advisable to not exceed the amounts recommended by experts in the field [143]. In the NO_3_^−^/NO_2_^−^ couple, the nitrite ion represents the greatest toxicological problem. In fact, being particularly reactive chemically, it can react with many functional groups, act as a reducing agent, or oxidize many reduced substrates [144]. In humans, nitrates in the digestive tract can be transformed into nitrite by the bacterial nitrate-reductase enzyme, generating robust toxicity phenomena due to the low pH. In the intestine, nitrates are responsible for the movement of chlorine ions and, at the same time, the elimination of sodium ions, which leads to a decrease in extracellular space. Finally, nitrates also appear to be responsible for thyroid dysfunction, nutritional effects, and reproduction [145,146]. In general, as already stated, the current acceptable daily intake for nitrites is 0.07 mg per kilogram of body weight per day, while that for nitrates is 3.7 mg/kg body weight/day.

### Main Plant Products Rich in Nitrates 

Beetroot (*Beta vulgaris*) belongs to the *Amaranthaceae* family and is cultivated all over the world. However, it does prefer the subtropical and tropical climates of Africa, Asia, and the Mediterranean countries [147]. Its roots contain numerous minerals, (K, P, Na, Mg, Cu, Ca, Zn, and Mg), vitamins, and phytochemicals (such as polyphenols and carotenoids). Today, beetroot is regularly consumed as part of a healthy diet and is also commonly used for the production of a food-coloring agent known as E162 [148]. Beetroot is rich in different bioactive compounds that can provide health benefits. In particular, it is helpful in treating disorders characterized by oxidative stress, chronic inflammation, endothelial dysfunction, and cognition. For this reason, the consumption of beetroot is especially recommended in regard to treating hypertension, type 2 diabetes, and dementia [149,150,151,152]. Beetroot juice (BTJ) triggers a continuous process responsible for a high concentration of nitrate and nitrite ions [153,154]. It increases nitrate concentration and promotes the production of NO, which can spread in vascular smooth muscle cells by binding to guanylyl cyclase, as well as allowing the production of c-GMP. This second messenger activates protein kinase G, thereby modulating smooth muscles and inducing relaxation [155]. Although there is no evidence of the association between a regular intake of beetroot and hypertensive patients, its consumption is considered a complementary and alternative strategy in hypertension. In fact, it has been proposed that the high concentration of inorganic nitrates that are contained in beetroot can compensate for the amount of NO reduced during hypertension, thereby assisting in regulating blood pressure [156]. The main results were provided by trials conducted between 2009 and 2017, in which the effect of beetroot juice on both physical performance and pressure levels was investigated [157,158,159]. The daily doses of beetroot juice consumed ranged from 70 to 500 mL, providing varying doses of NO_3_^−^ from 316 to 860 mg/100 mL of beetroot juice. The participants were aged between 21 and 67 years and the duration of the tests varied from 2 to 56 days. Overall, the results showed that groups ingesting beetroot juice had better results in physical exercise competitions, as well as lower pressure values, than the control group. Further, these differences appeared more marked in cases of prolonged treatment [160,161]. However, beetroot not only contains nitrate, but also many phytochemical compounds that can have beneficial health effects. These compounds include phenolic acids, ascorbic acid, flavonoids, carotenoids, and a group of highly bioactive pigments known as betalains [162,163]. In vitro and in vivo investigations have indicated that betalains possess antioxidant and anti-inflammatory capacities [164]; in fact, beetroot treatment is recommended in clinical conditions characterized by oxidative stress and chronic inflammation, such as in arthritis, liver disease, and cancer [165,166,167].

Spinach (*Spinacia oleracea* L.) belongs to the *Amaranthaceae* family and is an economically significant leafy vegetable grown worldwide [168]. To date, spinach is considered one of the most nutritious vegetables thanks to the presence of substances beneficial to human health, including vitamins A, E, C, K, folic acid, oxalic acid, and lutein, as well as minerals such as potassium, calcium, phosphorus, iron, magnesium, and manganese. Moreover, spinach also contains polyphenols, most notably lutein, zeaxanthin, and β-carotene [169]. The composition of spinach is responsible for many beneficial properties, including antioxidant and anti-inflammatory activities, protective effects against DNA oxidation, anticancer action, and defensive properties against atrial stiffness as well as intrahepatic stones and gallstones [170,171,172,173,174]. In recent years, numerous data have been published showing that dietary nitrate supplementation was able to induce a significant influence on arterial hemodynamics by means of nitric oxide supplementation [175,176]. In particular, vegetable-rich dietary models—i.e., featuring green leafy vegetables such as the Mediterranean diet or other dietary approaches against hypertension (DASH)—are associated with a decrease in the risk of cardiovascular disease, as well as in the reduction in arterial pressure and arterial stiffness [177,178]. Due to the fact that spinach is a source of high dietary nitrates (˃250 mg NO_3_/100 g), specific studies on arterial stiffness as well as central and peripheral blood pressure were conducted in healthy patients who were fed for 7 days with a daily high-spinach intake (~845 mg). The results showed that the administration of a high nitrate content reduced postprandial arterial rigidity and blood pressure values. Moreover, these results were even maintained after at least one week of continuous integration [179]. 

Lettuce (*Lactuca sativa* L.) belongs to the *Asteraceae* family and its origins coincide with the Mediterranean region. In general, the family of the Asteraceae is extremely vast, characterized by about 23,000–30,000 species [180]. The composition of lettuce shows a very high water content (94–95%), which is what makes this vegetable a low-calorie food. In addition, lettuce contains minerals, vitamins, glycosylated flavonoids, phenolic acids, tocopherols, carotenoids, polyphenols, and sesquiterpene lactones [181,182,183,184]. The intake of lettuce is the basis of various healthy effects. In fact, this plant exerts beneficial actions that aid in reducing the risk of the onset of chronic diseases (i.e., cardiovascular disorders, diabetes, cancer, and neurodegenerative diseases), as well as oxidative and inflammatory damage [185,186,187]. Lettuce ingestion has recently been shown to significantly increase both nitrates and total nitrites in saliva and urine samples when compared to a control group [188]. Furthermore, as reported in the literature, they contain the most well-known compounds that are part of the common diet [189]. 

Polyphenols are secondary metabolites of plants and are represented by a large number of compounds (about 10,000) that exercise protective activities for human health. In plants, polyphenols—in addition to being involved in the maintenance of the organoleptic properties of plants and food—operate certain defensive strategies, thereby acting as protectors against multiple types of stress, including pathogens, oxygen and nitrogen species, parasites and plant predators, UV light, and oxygen and nitrogen species, among others. In contrast, the main properties exercised in humans include those of the antioxidative, anti-inflammatory, anti-cancer, and cardio-protective variety [190,191,192,193,194,195,196]. A successful strategy of administering polyphenols is to be found not only in the prevention/treatment of various pathologies, but also in the slowing of the progression of pathologies and in promoting the healing process [197]. There is substantial epidemiological evidence that a diet rich in polyphenolic compounds protects against the development of several cardiovascular diseases [198,199]. In addition, these natural compounds increase endothelial function [200,201], reduce blood pressure [202] and arterial stiffness [203], and inhibit platelet aggregation [204], thereby suggesting that they can restore the correct concentration of NO as well as trigger vascular protection [205,206]. Finally, polyphenols are able also to promote the reduction in nitrates/nitrites to NO, thereby intervening in the modulation of post-translational reactions of nitric oxide. In this case, therefore, the local and systemic effects of NO are entrusted to polyphenols [207]. Among the countless polyphenols that exercise vascular protection, we will look deeper, in particular, at those contained in citrus. Bergamot (*Citrus bergamia*, Risso et Poiteau), is a citrus belonging to the *Rutaceae* family and to the genus *Citrus* that grows in southern Italy. Those of the best quality are found in the province of Reggio Calabria, Italy [208,209]. Bergamot, similar to other citrus fruits, is mainly rich in flavonoids and has beneficial properties for human health, including antioxidant and anti-inflammatory properties [210,211]. In addition, other activities are well-known, including the modulation of immunological, anti-cholesterolemic, and cardioprotective properties [212,213]. A fraction of bergamot, which is obtained both in its juice and albedo, is the polyphenolic fraction of bergamot (BPF). This fraction is enriched with polyphenols, reaching a concentration of 40%. The main components are naringin, neohesperidin, and neoerythocyte, as well as glycosylated polyphenols such as melitidine and bruteridine [214]. Recent data have shown that bergamot polyphenols are also able to exert a reduction in the levels of glucose, cholesterol, serum triglycerides, and systemic inflammation, as well as an improvement in endothelial function [134,215,216,217,218,219]. Due to the fact that naringin, hesperidin, and neoeriocitrin are all closely related to the modulation of nitrates/nitrites/NO [220,221,222,223,224,225], it is reasonable to assume that the abovementioned BPF can be considered a nutritional supplement to ensure the correct concentration of NO. This hypothesis arises from total speculation, though one that is justified by the theoretical study of the current literature, and for this reason it is an indispensable idea that deserves to be investigated in numerous (and appropriate) in vitro, in vivo, and clinical studies. In conclusion, it is possible to say that nitrates ingested in the diet can turn into nitrites and then NO, avoiding in most cases the formation of harmful N-nitroso compounds. On the other hand, the intake of nitrites, present in some processed foods, can generate nitroso-amines, which are harmful and carcinogenic. The meat industry uses nitrates/nitrites as additives during processing, as these compounds produce beneficial outcomes related to antioxidant effects, color enhancement, their antimicrobial role, and achievement of the typical flavor of sausages [226]. However, 10–20% of the originally added nitrite remains in the final product, is ingested in the diet, and added to the other endogenous and exogenous sources of these compounds. The International Agency for Research on Cancer (IARC) recently stated that nitrites can be considered carcinogens and have increased cancer rates [60]. In general, nitrites, and the NO derived from them, perform important physiological activities related to blood pressure and immune response. However, they can be considered dangerous under oxidative stress conditions as they can be converted into reactive nitrogen species (RNS) [227,228]. An increase in RNS promotes the occurrence of many acute and chronic diseases [229]. Nitrosative stress levels are mainly related to the time and concentration of exposure to RNS as well as to the ability of cellular antioxidants to remove these compounds [230]. Its uncontrolled intracellular presence produces significant toxicity as it interacts with biomolecules including proteins, DNA, lipids, and carbohydrates [231]. RNS-related toxic events can culminate in cytotoxicity, genotoxicity, and carcinogenesis. A representation of the protective and harmful role of nitrogen compounds ingested through the diet is shown in Figure 5.

As mentioned above, nitric oxide (NO) plays a particularly key role in the cardiovascular system [232]. In fact, the reduced bioavailability of NO, both from reduced production and increased consumption, has been associated with endothelial dysfunction and the onset of numerous cardiovascular diseases (CVDs), including atherosclerosis, ischemia–reperfusion injury, hypertension, and diabetes [233]. To date, it is known that restoration of the NO supply is able to positively affect CVD. Moreover, it has recently been accepted that the production of NO can also take place with the reduction of nitrate and nitrite inorganic anions [234]. These compounds are also present in the diet, particularly nitrate in green leafy vegetables [235]. It has been shown that intake of nitrite and nitrate (in the correct concentrations) has the ability to generate NO-like effects in the cardiovascular system. In particular, diets with a higher nitrate intake such as the Mediterranean diet are recommended for myocardial infarction and treatment of hypertension, peripheral artery disease, and pulmonary hypertension. Historically, nitrate has been used to relieve the symptom of chest pain in a heart attack by placing it under the tongue. At the same time, inorganic nitrite, in the form of potassium nitrite, has been used as a treatment for angina since 1883. Sodium nitrite in injectable form has become prevalent for the treatment of angina and hypertension [236]. After initial enthusiasm, these drugs were prescribed and used indiscriminately, generating a substantial number of side effects including hypotension and lethal methemoglobinemia [237]. It is precisely for this reason that nitrites and nitrates must be used consciously, without exceeding the permitted and previously indicated limits.

## 6. Discussion and Conclusions

GTN is widely used for the treatment of acute heart disease, including acute heart failure and coronary heart disease [238,239]. Its beneficial effects are associated with its ability to dilate arteries and veins, as well as in its ability to reduce the work of the myocardium. The mechanism of action underlying vasodilation is found in the release of NO, which is in response to the intracellular biotransformation of GTN by the enzyme ALDH2. In fact, mice that are deficient in mitochondrial ALDH2 were demonstrated to show impaired relaxation following administration with GTN [240]. Furthermore, long-term administration of organic nitrates causes tolerance and subsequent endothelial dysfunction. It is important to note that the reduction in vascular expression of ALDH2, induced by increased production of ROS, induces the early onset of tolerance. This is even after treatment with GTN for 48 h [115]. Tolerance to nitrates also involves soluble guanylyl cyclase (sGC), a heterodimeric enzyme that represents an important receptor for NO, which by binding to the heme of the H-NOX domain causes conformational changes, following which the enzyme is activated and catalyzes the conversion of GTP to cGMP. Furthermore, NO can form a covalent bond with the SH group of a Cys residue (S-nitrosylation), in particular the C122 present in the H-NOX domain, responsible for negative regulation by NO, determining the desensitization of sGC. In fact, following S-nitrosylation, conformational changes occur in the vicinity of the S-nitrosylated C122 residue that can prevent heme incorporation or the NO-activation of sGC, thus becoming mechanisms responsible for the desensitization of sGC. This is due to the aberrant NO responsiveness and is one of the mechanisms responsible for the onset of nitrate tolerance [241]. This is in addition to an altered biotransformation of organic nitrates to NO and a significant increase in the production of reactive superoxide species in the vascular system, which is responsible for a further inactivation of ALDH2 [242]. To date, the scientific community agrees that it would be advisable to remove nitrates from the human diet, as this would increase the concentration of NO. This would also help to avoid treatments of GTN being required, the induction of tolerance, the required increased GTN concentrations that would be needed otherwise, and the possible inhibition/alteration of the ALDH2 enzyme (which aggravates the issue with a positive feedback mechanism) [243,244]. In the context of diet, nitrates are mainly contained in fresh vegetables; having said this, how much is present depends on the specific vegetable [245]. It is also important to point out that the content of these compounds varies in relation to the parts of the plant in question. For example, Hord et al. have indicated that the content of NO_3_^−^ in the different organs of plants can be classified from the highest to the lowest as petiole > leaf > stem > root > tuber bulb > fruit > seed [136]. Beet, arugula, and spinach are the richest vegetables in NO_3_^−^ and have shown better effects on cardiovascular performance, reducing blood pressure and improving vascular function [246]. However, it is unthinkable that the introduction of nitrates into the diet would solve NO deficiency, as the change in the content of these compounds in the same vegetable occurs for several reasons. Among them, it is important to remember the conditions of growth, the season, the temperature, the meteorological conditions, the luminosity, the humidity, the age of the plants, the composition of the soil, the pH, the number of applications of fertilizers to increase the fertility of the soil, and the conditions of conservation of the collected plants [247,248]. For example, instant freezing induces a slight reduction in the amount of nitrate, within a period of seven days; boiling is able to reduce nitrate levels in vegetables by 47–59%; and finally, the process of frying in soybean oil increased nitrate content by 159–309% [249]. These innumerable variables constitute the main criticality of this topic. An interesting perspective for future development would be to be able to standardize the protocols of extraction of nitrates from different plants, reducing or eliminating all these changing conditions. Another important prospect could be to avoid the intake of nitrites that can turn into toxic nitrosamines—such as those contained in processed foods including meat and sausages—as these can aggravate, as already mentioned, human health [250]. To date, scientists have concluded that nitrates are both harmful and healthy, thus constituting the “paradox of nitrates”: to increase their beneficial role, it would be advisable to consume fresh and various vegetables and roots, which are rich in these compounds.

## Figures and Tables

**Figure 1 ijms-23-15454-f001:**
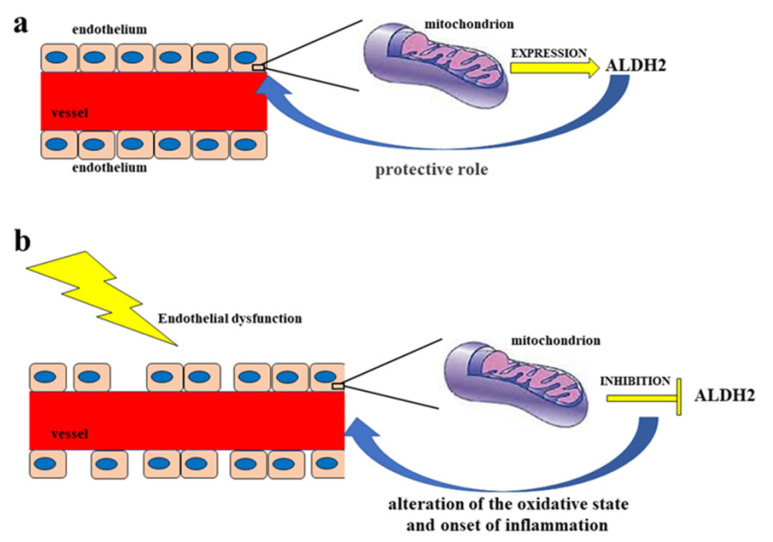
Double control of the endothelium regarding ALDH2. Diagram (**a**) represents a functioning endothelium with a proper mitochondrial expression of the enzyme ALDH2, which has a protective action on the endothelium itself. On the other hand, in diagram (**b**) the dysfunctional endothelium determines the inhibition of the mitochondrial synthesis of ALDH2, with a consequent negative effect on the endothelium, which is undergoing an alteration of the oxidative state and the onset of inflammation.

**Figure 2 ijms-23-15454-f002:**
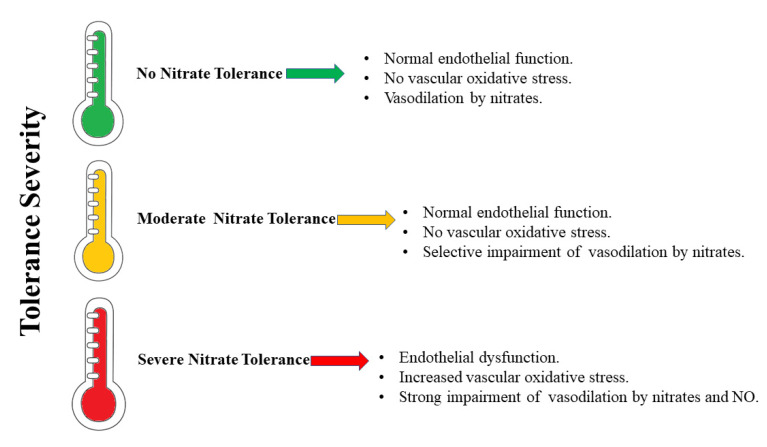
The severity of nitrate tolerance and its effects.

**Figure 3 ijms-23-15454-f003:**
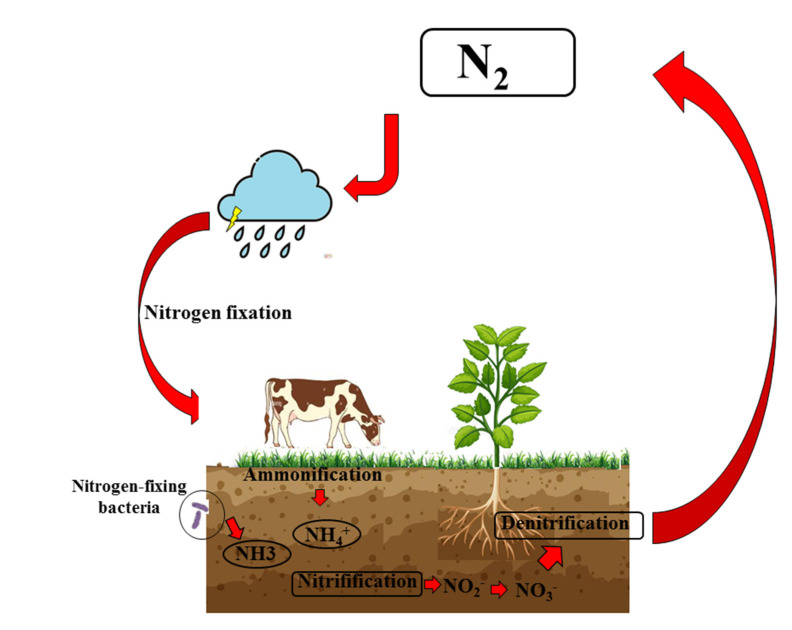
Biogeochemical nitrogen cycle. The difficult assimilation of N2 makes it necessary to organify nitrogen. The first step is known as “nitrogen fixation”, in which N_2_ is made available as ammonia (NH_3_). A subsequent process known as “ammonification”, carried out by fungi and various soil bacteria, is responsible for the formation of the ammonium ion (NH_4_^+^). Subsequently, this ion can be nitrified by some bacteria free in the soil, which generate nitrites and nitrates. Nitrates represent the most bioavailable form of nitrogen for plants, and they are absorbed at a radical level, transformed into vegetable proteins, and used as a source of nitrogen for the whole trophic network. The nitrogen cycle ends with the “denitrification” phase, in which some bacterial species again lead to the formation of N_2_.

**Figure 4 ijms-23-15454-f004:**
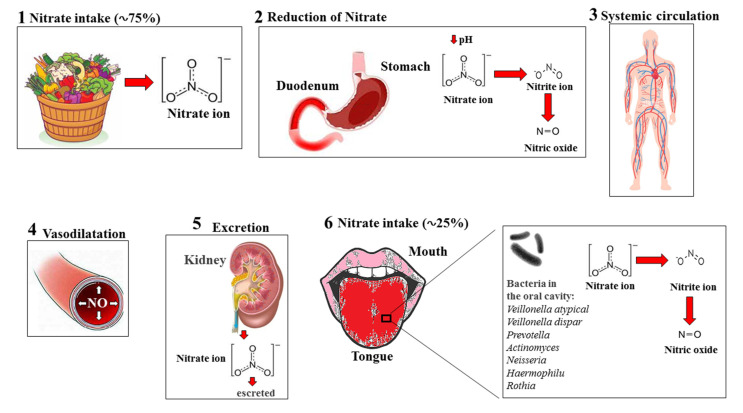
Processing of exogenous nitrates from food. Following exogenous dietary intake (**1**), most NO_3_^-^ ions are absorbed by the stomach and duodenum. Furthermore, they undergo reduction reactions, stimulated by low pH values in these regions, which also leads to the formation of NO_2_^−^ and NO (**2**). Subsequently, the nitrogen compounds are absorbed by systemic circulation (**3**), where there follows an increase in NO production and vasodilation (**4**). Finally, NO_3_^−^ is normally excreted in the urine by the kidneys (**5**). In addition, a component of nitrates, concentrated in the salivary glands, undergoes a reduction induced by bacteria in the oral cavity and is transformed into nitrite and NO (**6**).

**Figure 5 ijms-23-15454-f005:**
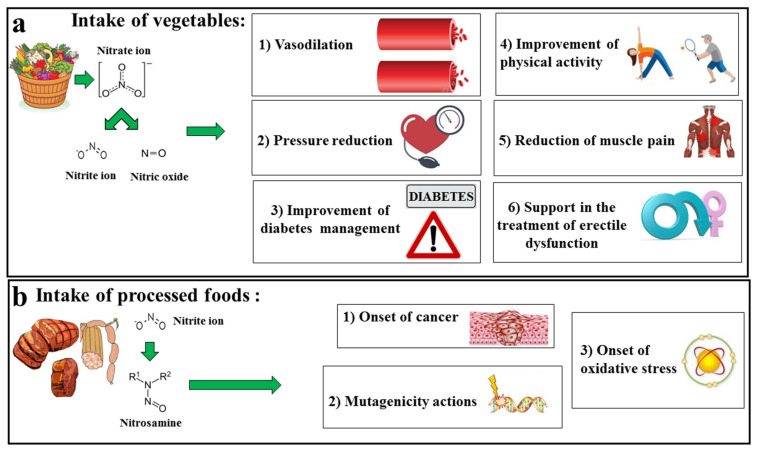
The protective and harmful roles of nitrogen compounds ingested through the diet. In diagram (**a**) the intake of plant products, which offer a large amount of nitrates, is shown. Subsequently, nitrates are transformed into nitrites and NO. The increased concentration of NO has beneficial effects on health, including vasodilation, reduction in blood pressure, better control of diabetes, better physical activities, reduction in muscle pain, and an improvement in the treatment of erectile dysfunction. In contrast, diagram (**b**) highlights the main effects from the intake of processed foods. The nitrites present in these foods can form toxic nitrosamines that are able to facilitate the onset of cancer, induce mutagenicity actions, alter biological macromolecules, and promote oxidative stress.

## Data Availability

Not applicable.

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
