# Peer review of "The Generation of Nitric Oxide from Aldehyde Dehydrogenase-2: The Role of Dietary Nitrates and Their Implication in Cardiovascular Disease Management"

_ijms, 2022, doi:10.3390/ijms232415454_

Round 1

Reviewer 1 Report

Review of a Review article:

«The Generation of Nitric Oxide From Aldehyde Dehydrogenase-2: Role of Dietary Nitrates and Implication in Cardiovascular Disease Management»

by Jessica Maiuolo, Francesca Oppedisano, Cristina Carresi, Micaela Gliozzi, Vincenzo Musolino, Roberta Macrì, Federica Scarano, Annarita Coppoletta, Antonio Cardamone, Francesca Bosco, Rocco Mollace, Carolina Muscoli, Ernesto Palma, and Vincenzo Mollace.

In International Journal of Molecular Sciences (ISSN 1422-0067).

Round 1

This work demonstrated a possible way of nitrates and nitrites transformation in the blood and tissues to form NO. Review aiming at investigating which vegetables should be taken and which foods are best avoided, based on current knowledge.

The article was presented in a weak-structured manner and weak scientific purpose and soundness in this version. Unfortunately, several statements within have weak evidence and arguments that should be expanded with more valuable proof. Therefore, the referee suggested that the manuscript should be considerably improved in a major revision. The following is a list of specific concerns.

1.     Introduction. In the introduction section, it is necessary to clearly define the range of issues considered in the review and the time period. Here it is also required to note whether there are reviews on similar topics in the world literature of the last ten years. List them and indicate the principal differences between this review and the already available ones.

-       The purpose of the Introduction is to attract the reader of IJMS by what he can find here in contrast to other similar publications.

-       The purpose of this review should be reconsidered and restructured (in introduction and in abstract section).

2.     The review's conclusion should be a detailed conclusion about the state of the field of science to date, with a brief description of achievements, shortcomings, and prospects for future development.

3.     Graphical abstract should be added.

4.     The primary function of NOX proteins should be more specified.

-       https://www.nature.com/articles/cmi201489#Sec14

5.     Are there any works concerning a variety of reaction pathway of nitrogen form? Appropriate illustration should be added in the future versions of this review in any suitable section.

6.     What concentrations are we talking about for nitrate and nitrite metabolites? What methods are suitable for their estimation in vivo? Please, provide brief description.

7.     Table 1 can be expanded and categorized. This edition has very little information.

Style guide issues

·       Line 30: nitrate (NO3-) and nitrite (NO2-) should be nitrate (NO3) and nitrite (NO2) with subscript and superscript characters, and it should be minus (–), not dash (-), as well;

·       Line 70: It should be minus (–), not dash (-);

·       Line 277 (Line 293, as well): Extremely unfortunate abbreviation («ischemia reperfusion injury = (IR injury)») that confuses the reader with IR radiation, use the established abbreviation Ischaemia-Reperfusion injury (IRI);

·       Line 477, 504 etc. Incorrect use of NO3, please use NO3 or nitrate;

·       Line 563: According to IUPAC nomenclature nitroso-amines should be spelled as nitrosamines or N-Nitroso amines (https://goldbook.iupac.org/terms/view/N04167).

English spelling should be double-checked.

Reviewer 2 Report

Hi,

This review manuscript is admirable collection of literature and information on ALDH2 and its role in CVD management using nitrates. My comments are listed below: -

1) In section 5.1, Table 1 is taken from a paper by Chung et al 2003. However, there are many other such studies (in recent years as well) that have published nitrate contents in vegetables. Therefore, using all those published data, a consensus table with statistical numbers should be generated for this review.

2) In section 5.1, authors talk about beetroot and bergamot but their nitrate contents are not listed in the table.  Also, Table 1 shows radish as a third most nitrate containing vegetable, but it's not discussed in this section as well.

3) This review has a title of ALDH2 generating NO, particularly from GTN but its proposed mechanism is not explained (for instance in article by Lang et al, JBC 2012).

4) While explaining nitrate tolerance effect on ALDH2, a brief mention of sGC desensitization by NO will be helpful (as explained by Kumar et al. Biochemistry 2013). 

5) Figure 2 attempts to explain the tolerance severity qualitatively. It would be more convincing representation if quantitative tolerance scale is presented as well.

6) Lines 451-454 describe about harmfulness of high concentrations of nitrates/nitrites/NO. It is advised to specify the exact numbers of recommendation values for a better clarity to the audience.

7) Figure 4b illustrates a scenario of processed food and its link to cancer, therefore it would be important to shed some deeper insights on ALDH2, nitrates and carcinogenicity in more details as well. 

8) Also, in addition to discussion and conclusion section, there should be a recommendation section in which intake of nitrates and veggies for CVD management are described in brief (as listed by health organizations around the globe).

9) There are some grammatical/typographic errors, some are listed below: -

a) Line 217 - gaunilic to guanylate

b) Line 83 - 1,0-1,8 to 1.0-1.8

Thank you